# Learning Counterfactual Interventions for Self-Supervised Motion Estimation

## Abstract

A major challenge in self-supervised learning from visual inputs is extracting information from the learned representations to an explicit and usable form. This is most commonly done by learning readout layers with supervision or using highly specialized heuristics. This is challenging primarily because the self-supervised pretext tasks and the downstream tasks that extract information are not tightly connected in a principled manner—improving the former does not guarantee improvements in the latter. The recently proposed counterfactual world modeling paradigm aims to address this challenge through a masked next frame predictor base model which enables simple counterfactual extraction procedures for extracting optical flow, segments and depth. In this work, we take the next step and parameterize and optimize the counterfactual extraction of optical flow by solving the same simple next frame prediction task as the base model. Our approach achieves state of the art performance for motion estimation on real-world videos while requiring no labeled data. This work sets the foundation for future methods on improving the extraction of more complex visual structures like segments and depth with high accuracy.

## 1 Introduction

Accurately estimating visual properties of the physical world from visual inputs is an essential capability for building intelligent embodied agents. Recently there has been significant progress in achieving this goal using video data, as evidenced by developments in video vision language models (Wang et al., 2024a), generative video models (OpenAI, 2024; Blattmann et al., 2023; Yang et al., 2024), and spatiotemporal self-supervised learning models (Bardes et al., 2024; Feichtenhofer et al., 2022; Qian et al., 2021). These powerful models learn an implicit representation of a wide variety of visual properties such as object motion, shape, material properties, and semantic relationships. However, for these abilities to be practically useful, they require a means for explicitly extracting such properties from the representation.

There are two main existing paradigms for explicitly extracting visual properties from representations: supervised and heuristic. In the supervised paradigm, base representations are fine-tuned to support read-out layers using labeled datasets (Bardes et al., 2024; Feichtenhofer et al., 2021). This is suboptimal because it requires costly labeled data for each task of interest. In contrast, heuristic approaches exploit emergent feature-level correlations by applying various nearest neighbor or clustering procedures (Jabri et al., 2020; Bian et al., 2022; Amir et al., 2022), or use strong task-specific regularizations like smoothness (Jiang et al., 2024; Stone et al., 2021). The heuristic approach is limited because the relationship between the desired property to be extracted and the heuristic criterion is often indirect and only truly valid for a subset of data inputs. This makes it difficult to improve heuristic methods in a principled way—getting a better loss on the pre-training or pretext task is not guaranteed to yield better extractions.

This begs the question: Is there a paradigm that enables explicit extraction that is both principled, in the sense that it is tightly connected with how the base model is trained, but that can also be improved without labeled data? The Counterfactual World Modeling (CWM) paradigm (Bear et al., 2023) (see Figure 1A and B) seeks to satisfy this requirement. It proposes a technique for self-supervised training on videos that enables the extraction of scene properties like optical flow, segmentation, and depth with simple generic procedures. The base model in CWM is a *sparse RGB-conditioned next*

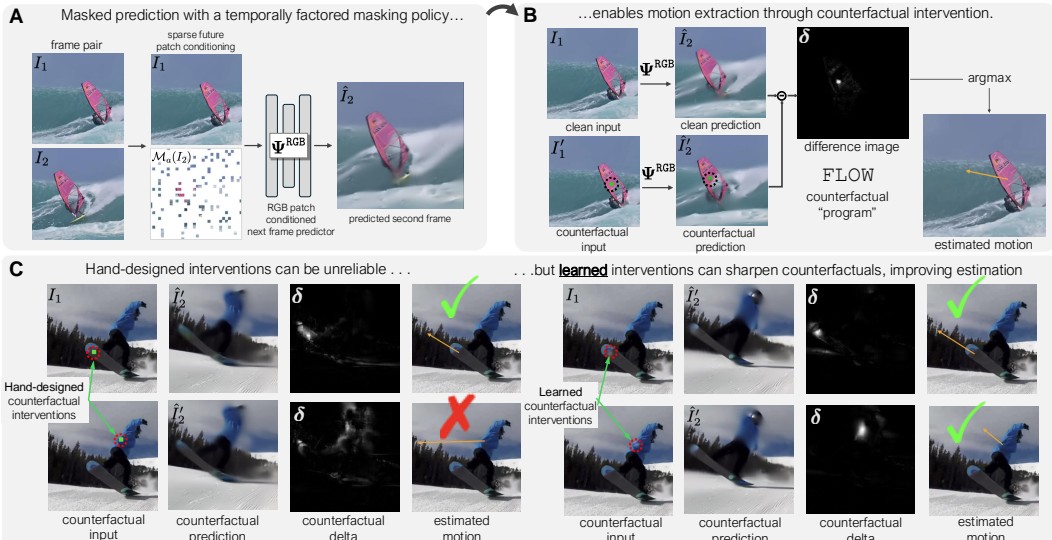

Figure 1: **(A)** Counterfactual world models (CWM) learn to predict the next frame with a temporally factored masking policy. **(B)** After training such a predictor, the motion of a point can be estimated using a simple counterfactual "program": the model predicts the next frame with and without a colored patch placed on the point, and the difference between the predictions reveals the estimated motion. **(C)** Hand-designed interventions are out of domain for the CWM predictor, causing inconsistent motion estimates. We propose a technique for learning to predict *learned interventions* without any labeled data, enabling state-of-the-art unsupervised object motion estimation.

*frame predictor* $\mathbf{\Psi}^{\text{RGB}}$, a two-frame masked autoencoder with a temporally factored masking policy. It learns to predict the pixels of the second frame based on the first frame and a small set of revealed patches of the second frame. To solve this task the model has to implicitly learn about the physical properties of objects and their dynamics. Scene properties are then explicitly extracted from from the base model through *simple counterfactual programs*. Counterfactual programs start with counterfactual interventions—simple changes to the predictor's inputs, such as placing a visual marker or moving an image patch, resulting in counterfactual predictions. Explicit properties are then derived by further processing based on the difference between the clean (or factual) and counterfactual predictions.

A key example of this concept is estimating optical flow. In the FLOW (Figure 1B) counterfactual program for extracting object or scene motion, the intervention is a distinctive perturbation placed on the point we want to track in the first frame. The RGB-conditioned predictor receives this perturbed input frame along with a clean input frame and makes a prediction with and without the intervention. FLOW then estimates motion by deriving where the predictor "carries" the perturbation by comparing the clean and counterfactual predictions. Because this FLOW counterfactual essentially provides an algorithmic definition for the "flow" concept, there is a principled direct connection between minimizing the loss of the base model and the accuracy of the extraction procedure.

Bear et al. (2023) showed that initially promising with this approach, using bright colored patches as the perturbation. These hand designed perturbations, however, can be unreliable, in part because they are out of domain for the RGB-conditioned base predictor, sometimes leading to spurious predictions and inconsistent motion estimation. To improve the connection with $\mathbf{\Psi}^{\text{RGB}}$ and improve extractions from FLOW, here we recast it as a differentiable program `diffFLOW` with learnable parameters by introducing a function that predicts the appearance of the markers used for intervention. We propose to optimize `diffFLOW`'s parameters by connecting its outputs to a flow-conditioned next frame predictor $\mathbf{\Psi}^{\text{FLOW}}$ and doing joint optimization. Forcing $\mathbf{\Psi}^{\text{FLOW}}$ to predict a future frame based on a present frame flow creates an information bottleneck which guarantees useful gradients for optimizing the parameters of `diffFLOW`. Through this approach, we are training an extraction procedure through the same unsupervised next-frame prediction task as the base predictor. We focus on optical flow because motion estimation is the most fundamental notion of visual correspondence, from which higher-order properties like shape, object segments, and dynamics can be derived.

We find that CWM with optimized counterfactual interventions outperforms state-of-the-art unsupervised motion estimation methods that are purposely built for this task (Stone et al., 2021; Jiang et al., 2024) when evaluated on a challenging real-world motion estimation benchmark (Doersch et al., 2022). Learning an optimized counterfactual intervention results in large performance improvements relative to fixed interventions, revealing a promising direction for future work to improve counterfactual extraction of other visual structures.

## 2 RELATED WORK

**Self-Supervised learning from video** Many prior works focus on developing self-supervised representation learning objectives by leveraging the inherent spatio-temporal structure in videos targeting downstream tasks like video action recognition or temporal correspondence. These methods can be broadly categorized into predictive and contrastive. Predictive techniques learn by making predictions about the temporal ordering of videos (Wei et al., 2018; Misra et al., 2016), predicting missing information for a target frame given a context frame in pixel space (Vondrick et al., 2018; Recasens et al., 2021), or in feature space (Bardes et al., 2023; 2024), or by following a spatio-temporal masked autoencoding paradigm in pixel (Tong et al., 2022; Wang et al., 2023a; Feichtenhofer et al., 2022) or feature (Wang et al., 2023b) space. Contrastive representations get trained by learning to encode temporally close frames (Feichtenhofer et al., 2021; Qian et al., 2021; Xu & Wang, 2021) or spatio-temporally close patches (Jabri et al., 2020; Bian et al., 2022; Li et al., 2019) with similar features. Counterfactual world models are another class of video predictive models that use a temporally-factored masking policy Bear et al. (2023) during training. Various vision structures can be extracted using a single pre-trained model by defining them as counterfactuals. This extraction process has parameters which need to be chosen by hand which leads to sub-optimal structure extractions. In this paper, we provide a recipe to improve the extraction procedure by designing the counterfactuls in a way that supports differentiable optimization through the pre-trained predictor.

**Self-supervised motion estimation** These works specifically focus on learning how to estimate short or long term motion in videos without any supervision. Some works are based on prior contrastive and predictive techniques (Bardes et al., 2023; Bian et al., 2022; Xu & Wang, 2021) or prior optical-flow methods (Stone et al., 2021; Jiang et al., 2024).

**Visual prompting** With the success of few-shot in-context learning methods for language prompt optimization in LLMs and VLMs, there has been increasing interest in understanding their ability to recognize visual prompts ( Nasiriany et al. (2024)) which offer the advantage of visually cueing the model with more granular control. Yang et al. (2023) does a comprehensive study on the ability of VLMs to understand a wide variety of visual prompts. Works such as Shtedritski et al. (2023) investigate whether visual prompt engineering can be used to extract meaningful predictions from VLMs. Counterfactual World Models ( Bear et al. (2023)) use a form of visual prompting via patch-level interventions that involve making modifications to the input patches to a masked video prediction model. These interventions can be used to extract meaningful structures. In this work, we explore whether the intervention can be optimized for better motion estimation using CWM, drawing parallels to prompt-optimiziation in VLMs.

**Self-supervised learning from images** Caron et al. (2021) introduce a method of label free distillaltion of ViTs (DINO), demonstrating that semantic segmentation emerges in the attention maps of Vision Transformers (ViTs) trained with a contrastive learning objective. Oquab et al. (2023) extend this approach by scaling up contrastive pre-training across larger datasets and model architectures, enhancing overall performance. Additionally, they find that other vision tasks, such as depth estimation, can be derived by training a linear probe on the model's frozen features with minimal data. A similar observation is reported in latent diffusion models such as Stable Diffusion Rombach et al. (2022), where attention maps facilitate zero-shot extraction of semantic segmentation information Tian et al. (2024). Recent works have also found that long-range dense point tracking in videos can be extracted using test-time optimization, leveraging DINO's pre-trained for learning correspondences Tumanyan et al. (2024).

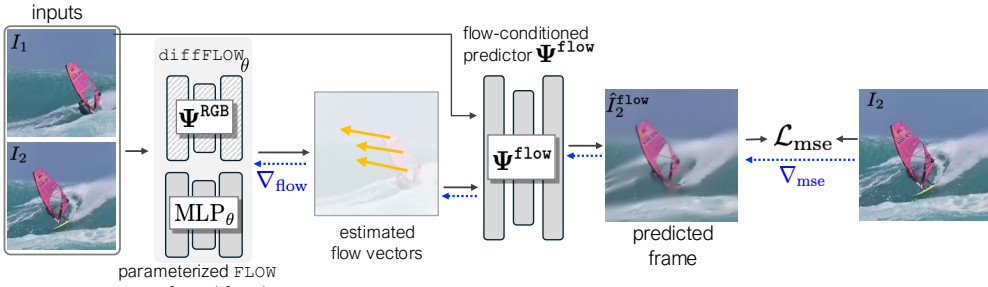

Figure 2: **Improving counterfactual extractions by next-frame prediction** The counterfactual program `diffFLOW` extracts motion from a frozen RGB-conditioned predictor $\mathbf{\Psi}^{\text{RGB}}$ through counterfactual intervention (details in Figure 3). Its parameters are trained using gradients from a flow-conditioned predictor $\mathbf{\Psi}^{\text{flow}}$ that is jointly trained to perform next-frame prediction. The $\mathbf{\Psi}^{\text{flow}}$ can only learn to predict future frames if given correct flow vectors. This explicit information bottleneck ensures useful gradients will get passed back to `diffFLOW`. This setup allows us to get better extractions from a pre-trained $\mathbf{\Psi}^{\text{RGB}}$ predictor by training another flow-conditoned predictor $\mathbf{\Psi}^{\text{flow}}$.

## 3 METHOD

In this section, we first revisit the counterfactual world modeling paradigm (Bear et al., 2023). We then present our approach for unsupervised learning of optimized counterfactual interventions. Last, we discuss several inference-time techniques that are essential for practical applications of the motion representations learned by CWM.

### 3.1 CWM: COUNTERFACTUAL WORLD MODELS

**RGB-Conditioned Next Frame Predictor** The first element of CWM is an RGB-conditioned next frame predictor $\mathbf{\Psi}^{\text{RGB}}$, consisting of an encoder $\mathbf{\Psi}^{\text{RGB}}_E$ and decoder $\mathbf{\Psi}^{\text{RGB}}_D$, similar to a Video-MAE (Tong et al., 2022), but trained with a temporally factored masking policy (see Figure 1A). Let $I_1, I_2 \in \mathbb{R}^{3 \times H \times W}$ be the two images in a video frame pair, and define $\mathcal{M}_\alpha$ as a masking function that randomly masks some fraction, $\alpha$, of patches in an image. Given a fully visible first frame $I_1$ and a partially visible second frame $\mathcal{M}_\alpha(I_2)$, $\mathbf{\Psi}^{\text{RGB}}$ is trained to predict $I_2$ by minimizing

$$\mathcal{L} = \text{MSE}(\hat{I}_2, I_2) \quad \text{where} \quad \hat{I}_2 = \mathbf{\Psi}^{\text{RGB}}\big(I_1, \mathcal{M}_\alpha(I_2)\big). \tag{1}$$

Setting $\alpha = 0.9$ creates a temporally factored masking policy. By predicting the second frame pixels given a full first frame and some visible patches of the second frame, $\mathbf{\Psi}^{\text{RGB}}$ is forced to learn what underlying scene transformations can explain what is revealed by the few visible patches.

**Counterfactual Interventions for Structure Extraction** The base predictor has a strong dependence on the appearance and position of the revealed patches from $I_1$ and $I_2$. This allows for extracting visual structure through applying counterfactual interventions: small changes to the appearance or the position of visible patches. By measuring the predictor's response to these counterfactuals, we can easily extract useful information like object motion, segments or shape from its representation. For the specific case of motion estimation, as shown in Figure 1 for the `FLOW` procedure, we can place a colored patch on a moving object and determine its motion by finding its location in the predicted frame. Formally, let $\mathcal{C} : (I, p) \mapsto I'$ be a counterfactual intervention function that takes an image $I$ and places a colored patch at pixel location $p = (u, v) \in [0, H) \times [0, W)$ to output the counterfactual input $I'$. To track a pixel $p_1$ we first get second frame predictions with and without the counterfactual intervention

$$\hat{I}'_2 = \mathbf{\Psi}^{\text{RGB}}\big(I'_1, \mathcal{M}_\alpha(I_2)\big) = \mathbf{\Psi}^{\text{RGB}}\big(\mathcal{C}(I_1, p_1), \mathcal{M}_\alpha(I_2)\big), \qquad \hat{I}_2 = \mathbf{\Psi}^{\text{RGB}}\big(I_1, \mathcal{M}_\alpha(I_2)\big). \tag{2}$$

Subtracting these two predicted frames and taking an $L_1$-norm across the color channels produces the difference image, $\boldsymbol{\delta} = |\hat{I}'_2 - \hat{I}_2|^c_1$ (where the superscript $c$ indicates the $L_1$-norm is only over the channel dimension, so that $\boldsymbol{\delta}$ retains its $H$ and $W$ dimensions). In turn, we retrieve the predicted pixel location $\hat{p}_2$ by finding the peak in the difference image: $\hat{p}_2 = (\hat{u}_2, \hat{v}_2) = \arg\max_{(u,v)} \boldsymbol{\delta}$. The concept of extracting visual structure through counterfactual intervention of a generic base predictor

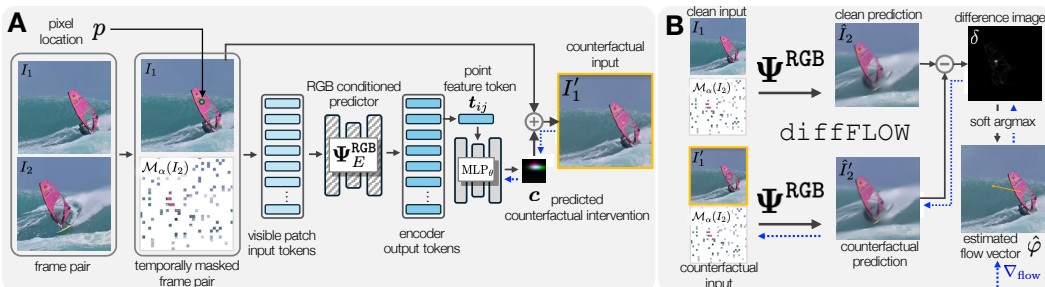

Figure 3: **Parameterizing counterfactual extraction into a function `diffFLOW`**: $(I_1, I_2, p_1) \mapsto \hat{\varphi}$. Building on a pre-trained RGB-conditioned predictor $\mathbf{\Psi}^{\text{RGB}}$, we cast the counterfactual flow extraction procedure as a feedforward differentiable function `diffFLOW` that can predict a forward motion vector for any pixel in an image pair. The parameters of the counterfactual intervention prediction $\text{MLP}_\theta$ are trained using gradients from an upstream flow-conditioned predictor $\mathbf{\Psi}^{\text{flow}}$.

is itself generic, allowing for extraction of other structures like segments and depth maps (for details on these extraction procedures see Bear et al. (2023)).

## 3.2 Optimizing Counterfactuals for Motion Estimation

**What makes a good counterfactual intervention?** The main requirement for a good counterfactual intervention is for it to cause meaningful changes in the outputs of the base predictor. While counterfactual interventions through colored patches can extract motion from $\mathbf{\Psi}_{\text{RGB}}$, as evidenced by Figure 1C., the appearance content of the patch can be suboptimal. While sometimes effective, a bright colored patch is *out of domain* for the base predictor. For a moving object, this results in failure cases like not appearing on the object in the second frame, not moving with the object, or unwanted artifacts in the prediction. All of these lead to noisy difference images and incorrect motion estimations. Can the appearance of the counterfactual intervention be optimized to avoid these failures and improve performance?

We propose a method for learning the parameters of a function that predicts the appearance of counterfactual interventions (see Figure 2) without using labeled data. We jointly train a counterfactual motion prediction function, `diffFLOW`, which estimates a set of flow vectors, and a flow-conditioned predictor, $\mathbf{\Psi}^{\text{flow}}$, which takes a single frame along with the flow vectors to predict the next frame. We improve `diffFLOW` by passing its outputs as inputs to $\mathbf{\Psi}^{\text{flow}}$ and training end-to-end using the RGB reconstruction loss of the predictions of $\mathbf{\Psi}^{\text{flow}}$. The information bottleneck at the input of $\mathbf{\Psi}^{\text{flow}}$, namely that it has no access to any RGB patches from the second frame $I_2$, supervises `diffFLOW` to produce accurate flow predictions, as $\mathbf{\Psi}^{\text{flow}}$ can only minimize its loss by incorporating this motion information.

### 3.2.1 A Function for Predicting Counterfactual Interventions

We re-formulate the motion extraction procedure from Section 3.1 to make it a parameterized differentiable function and introduce the functional form of a sum of colored Gaussians as a natural intervention class. Let `diffFLOW`: $(I_1, I_2, p_1) \mapsto \hat{\varphi}$ be a per-pixel motion estimation function with learnable parameters $\theta$ that takes an image pair $(I_1, I_2)$ and a pixel location $p_1 = (u_1, v_1)$ in $I_1$ and outputs the predicted flow $\hat{\varphi} = \hat{p}_2 - p_1 = (\hat{u}_2, \hat{v}_2) - (u_1, v_1) \in \mathbb{R}^2$. The function `diffFLOW` consists of multiple components: the counterfactual intervention function, $\mathcal{C}$, which now produces counterfactual inputs with Gaussian interventions instead of solid-color squares; the pre-trained, frozen, RGB-conditioned predictor, $\mathbf{\Psi}^{\text{RGB}}$; and a "softargmax module" to predict a pixel location using a differentiable approximation to the argmax function. Here, $\mathcal{C}$ uses the encoder $\mathbf{\Psi}_E^{\text{RGB}}$ from the RGB-conditioned predictor, and also contains a small MLP (with parameters $\theta$) that predicts the parameters of the Gaussian intervention.

**Gaussian Interventions** The RGB-conditioned encoder $\mathbf{\Psi}_E^{\text{RGB}}(I_1, \mathcal{M}_\alpha(I_2))$ outputs a sequence of feature tokens from its last transformer block. Given a pixel location $p_1 = (u_1, v_1)$, we find its corresponding token $\mathbf{t}$, and use it as an input to an MLP that outputs a parameter vector $\boldsymbol{\gamma} = \text{MLP}_\theta(\mathbf{t})$, which is used to compute the Gaussian intervention. Then, the counterfactual intervention function

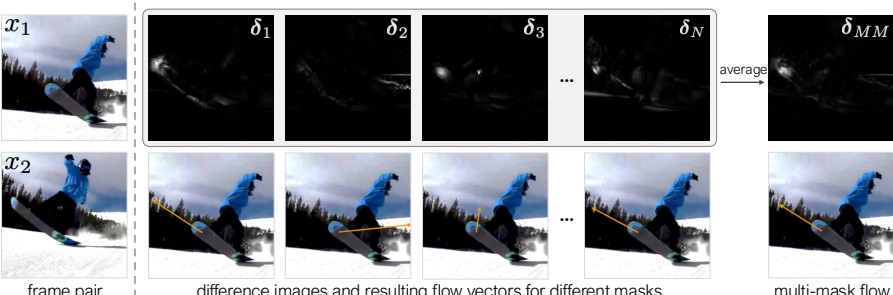

Figure 4: **Multi-mask inference results in reliable predictions:** Given a frame pair, we compute $N$ difference images $\boldsymbol{\delta}_1, \boldsymbol{\delta}_1, \ldots, \boldsymbol{\delta}_N$ with different random second frame masks $\mathcal{M}_\alpha(I_2)$. Observe the uncertainty in the difference images. Averaging the difference images into $\boldsymbol{\delta}_{MM}$ allows us to obtain a sharp peak and an accurate flow vector.

uses these parameters to compute one Gaussian PDF for each color channel to form the three-channel Gaussian intervention, which it then adds to $I_1$ to produce the counterfactual input $I_1'$.

We use Gaussians because this functional class presents a natural method of forming in-domain counterfactual inputs. Instead of solid-colored squares, which have a sharp cutoff and strongly saturated colors, Gaussians approach zero smoothly as distance from the mean increases, allowing for small colored bump-like interventions, which smoothly blend into the input image. We then compute the second frame prediction with and without the counterfactual intervention as in equation 2, and use these to compute the difference image $\boldsymbol{\delta}$. Because diffFLOW needs to be differentiable, we use a softargmax over $\boldsymbol{\delta}$.

**Softargmax Module** We follow the softargmax formulation proposed in Wang et al. (2020a). Given a difference image, $\boldsymbol{\delta} = |\hat{I}_2' - \hat{I}_2|_1^c$, we first apply a temperature-scaled 2D softmax and then take the expectation according to that softmax to find the predicted second frame pixel location $\hat{p}_2 = \mathbb{E}_{p_2 \sim \text{softmax}(\boldsymbol{\delta}/\tau)}[p_2]$. For distributions with a fairly localised peak, this expectation is a differentiable approximation of argmax. The predicted flow is computed as $\hat{\varphi} = \hat{p}_2 - p_1$.

### 3.2.2 LEARNING TO PREDICT COUNTERFACTUAL INTERVENTIONS WITHOUT SUPERVISION

Given an image pair $(I_1, I_2)$, we estimate the motion for a set of pixels $\mathcal{P} = \{p_1^{(1)}, p_1^{(2)}, \ldots, p_1^{(n)}\}$ using diffFLOW, obtaining a set of estimated forward flow vectors $\hat{\mathcal{F}} = \{\hat{\varphi}^{(1)}, \hat{\varphi}^{(2)}, \ldots, \hat{\varphi}^{(n)}\}$. Let $\boldsymbol{\Psi}^{\text{flow}}: (I_1, \hat{\mathcal{F}}) \mapsto \hat{I}_2$ be a flow-conditioned next frame predictor with parameters $\psi$ that takes the first frame RGB input $I_1$ and predicts the next frame $\hat{I}_2$, conditioned on the flow input $\hat{\mathcal{F}}$. We jointly optimize $\theta$ and $\psi$, by minimizing an MSE next frame reconstruction loss $\min_{\theta, \psi} \mathcal{L}_{\text{MSE}}(\hat{I}_2, I_2)$.

### 3.3 MULTI-MASK INFERENCE FOR MANAGING UNCERTAINTY

The base predictor's reconstruction of the second frame is strongly conditioned by the small set of visible patches. Next frame prediction has high uncertainty: even with a few revealed patches, there are many valid ways to reconstruct the rest of the future frame. This can cause noisy extractions for two reasons. First: because the reconstructed pixels will not necessarily be the same across different random samplings of visible patches may vary as well—this is an issue, because there is only one correct answer. Second, a patch may be revealed near the location of the motion we are trying to predict, which causes the model to not predict the counterfactual intervention in the future frame.

We implement a *multi-mask* (MM) inference procedure (see Figure 4). For MM-$N$, we run $N$ forward passes of diffFLOW (with different randomly generated masks output from $\mathcal{M}_\alpha$) until the difference image computation, with each forward pass producing difference image $\boldsymbol{\delta}_n$. The final MM prediction is the peak in the average delta image $\boldsymbol{\delta}_{MM} = \frac{1}{N} \sum_{n=1}^{N} \boldsymbol{\delta}_n$. In Table 3, we find that this results in large performance improvements.

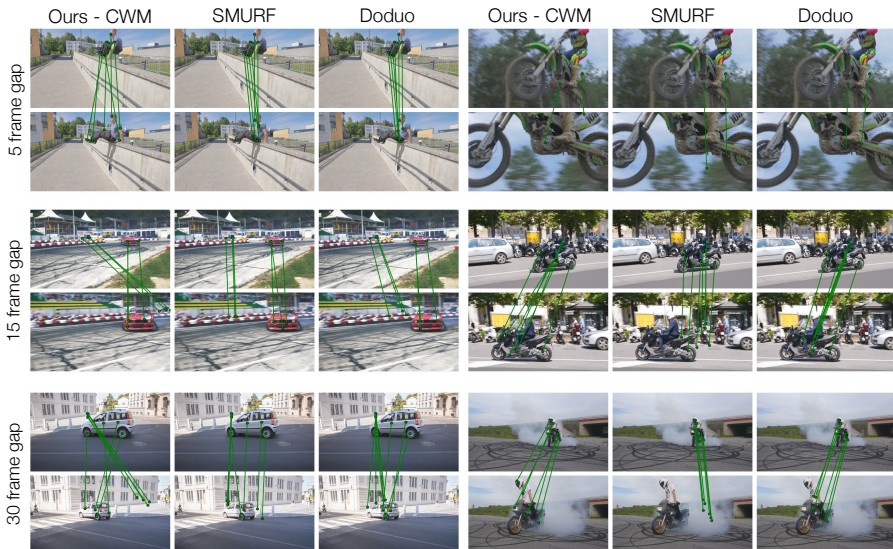

Figure 5: **Qualitative comparison with baselines on TAP-Vid DAVIS across various frame gaps** Compared with Doduo and SMURF, CWM can accurately estimate the motion of points on both foreground and background objects with high object and camera motion, as shown in the leftmost 15- and 30-frame gap examples. On foreground objects, CWM and Doduo both outperform SMURF in cases where SMURF loses the object entirely, while CWM more accurately predicts the locations of points on the extremities of these objects.

### 3.4 DISTILLING THE CWM MOTION REPRESENTATION

The multi-mask procedure is essential for high motion estimation accuracy but makes inference expensive—it takes up to 40 multi-mask iterations until the accuracy improvements start to diminish. To improve the practical utility of the CWM motion representation, we propose to distill it into an architecture purpose-built for optical flow estimation. We sparsely label a large video dataset with 5% visible patches per frame pair using our trained counterfactual motion prediction model. We use this to train the SEA-RAFT Wang et al. (2024b) model, which results in a fast and efficient motion estimation model trained without any labeled data.

### 3.5 IMPLEMENTATION DETAILS

The RGB-conditioned next frame predictor $\Psi^{\text{RGB}}$ is pre-trained with AdamW (Loshchilov & Hutter, 2019) using a learning rate of $1.5e - 4$ with cosine annealing after 40 epochs of linear warm up. We use a batch size of 1024 and train for a total of 800 epochs. The pre-trained predictor is then frozen and used to generate flow estimations through counterfactual interventions within `diffFLOW`. We use a similar optimization configuration to train our flow-conditioned next frame predictor, $\Psi^{\text{flow}}$. The model is trained with a batch size of 32 for 200 epochs. We set the temperature parameter $\tau$ for our soft-argmax module (Section 3.2.1) to $\frac{1}{200}$.

Both the RGB-conditioned predictor $\Psi^{\text{RGB}}$ and the flow-conditioned predictor $\Psi^{\text{flow}}$ are trained on Kinetics-400 (Kay et al., 2017). For pre-training $\Psi^{\text{RGB}}$, we sample frame pairs 150ms apart with center crop augmentation and resize to an input resolution of $256 \times 256$. We also fine-tune on $512 \times 512$ resolution with interpolated position embeddings as proposed by Dosovitskiy (2020).The flow-conditioned predictor $\Psi^{\text{flow}}$ was trained on frame pairs 500ms at $256 \times 256$ resolution. The purpose of this larger frame gap is to create a stronger dependence of $\Psi^{\text{flow}}$ on the quality of flow estimations from `diffFLOW`. Code will be released upon acceptance.

Table 1: **Quantitative comparison on TAP-Vid DAVIS—VFG** Our proposed approach, CWM with a learned counterfactual prompt prediction function, obtains state-of-the-art performance when compared with unsupervised baselines. $U^{\dagger}$ indicates self-supervised training with object masks.

| | Methods | Dataset | AD↓ | MD↓ | $< \delta_{avg}^{x} \uparrow$ |
|---|---|---|---|---|---|
| S | SEA-RAFT Wang et al. (2024b) | Sintel | 27.46 | 11.39 | 56.09 |
| | SEA-RAFT | KITTI | 20.19 | 6.99 | 59.01 |
| | SEA-RAFT | Spring | 23.75 | 12.79 | 51.44 |
| $U^{\dagger}$ | Doduo Jiang et al. (2024) | Youtube-VOS | 12.3 | - | 43.5 |
| U | Doduo w/o segment | Youtube-VOS | 13.0 | - | 39.8 |
| | SMURF Stone et al. (2021) | Sintel | 27.21 | 18.42 | 44.47 |
| | SMURF | KITTI | 41.48 | 33.25 | 34.54 |
| | SMURF | Chairs | 28.77 | 18.96 | 40.75 |
| | DINOv2 Oquab et al. (2023) | LVD-142M | 13.4 | - | 36.0 |
| U | CWM 512 MM-40 WBinit | Kinetics | **11.78** | **3.63** | **52.30** |
| | CWM 512 MM-40 | Kinetics | 12.45 | 4.62 | 47.50 |
| | CWM 256 MM-40 | Kinetics | 14.63 | 5.84 | 42.62 |
| U | CWM distilled into SEA-RAFT | Kinetics | 25.22 | 14.79 | 43.36 |

Table 2: **Quantitative comparison on TAP-Vid DAVIS CFG with a gap of $\Delta = 5$ frames.** Our proposed approach CWM, with a learned counterfactual prompt prediction function, obtains state-of-the-art performance when compared with unsupervised baselines purposely-made for optical flow. $U^{\dagger}$ indicates self-supervised training with object masks

| | Methods | Dataset | AD↓ | $< \delta_{avg}^{x} \uparrow$ |
|---|---|---|---|---|
| S | SEA-RAFT Wang et al. (2024b) | Sintel | 2.20 | 83.85 |
| | SEA-RAFT | KITTI | 1.61 | 84.98 |
| | SEA-RAFT | Spring | 2.12 | 79.45 |
| $U^{\dagger}$ | Doduo Jiang et al. (2024) | Youtube-VOS | 1.77 | 72.62 |
| U | SMURF Stone et al. (2021) | Sintel | 2.69 | **79.64** |
| | SMURF | KITTI | 4.54 | 71.27 |
| | SMURF | Chairs | 3.10 | 76.44 |
| U | CWM 512 MM-40 WBinit | Kinetics | **2.09** | 69.18 |
| | CWM 512 MM-40 | Kinetics | 2.41 | 59.24 |
| | CWM 256 MM-40 | Kinetics | 2.67 | 56.93 |
| U | CWM distilled into SEA-RAFT | Kinetics | 3.02 | 76.51 |

# 4 EXPERIMENTS

## 4.1 EVALUATION PROTOCOL

**TAP-Vid DAVIS—Variable Frame Gap (VFG)** We follow the procedure for motion estimation on real data from Doduo (Jiang et al., 2024) based on the TAP-Vid DAVIS dataset (Doersch et al., 2022) point tracking dataset. For each point in the 30 videos, we take the first frame where it appears as the source image, and every other frame where it is visible as the target image [1]. This is more challenging than optical flow estimation because it requires estimating the motion of a point under greater scene variability due to the variable frame gaps.

**TAP-Vid DAVIS—Constant Frame Gap (CFG)** We propose an additional protocol with fixed frame gaps. For each CFG evaluation run, we choose a frame gap $\Delta$ set to either 5, 10 or 15. For each TAP-Vid DAVIS video, we select all pairs of frames that are $\Delta$ apart, and compute metrics using all tracked points visible in both frames.

**Metrics** We use the average distance (AD) between the estimated pixel and ground truth pixel locations and $< \delta_{avg}^{x}$, which is the average percentage of predictions with an error of less than 1, 2, 4, 8 and 16 pixels. These metrics respectively measure the accuracy and precision of the predic-

---

[1]Unlike the original TAP-Vid (Doersch et al., 2022) procedure, but in line with the estimation done by Doduo (Jiang et al., 2024), we do not predict or evaluate the handling of occlusion

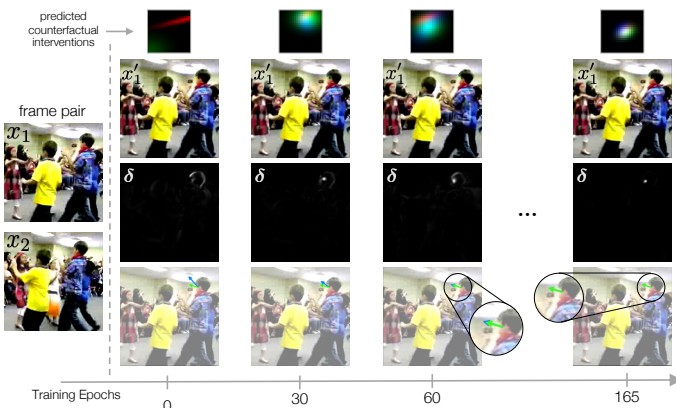

Figure 6: **Evolution of counterfactual interventions across training epochs:** We observe how the predicted counterfactual interventions change as the model trains. The intervention starts as a disjoint streak of colors and converges to a localised peak. This in turn increasingly concentrates the difference image $\delta$ and leads to better flow prediction. Green is the ground truth flow obtained from the TAP-Vid dataset, and blue is our model's prediction.

tions. Following Jiang et al. (2024), the metrics are computed after rescaling to a $256 \times 256$ image. To make sure each baseline is performing optimally, the input resolution is either the native video resolution or $256 \times 256$ depending on what results in the best performance.

**SMURF** is an unsupervised method specifically designed for optical flow estimation. This work tailors the RAFT (Teed & Deng, 2020) architecture so it can be trained using optical flow-specific heuristics losses like photometric loss and smoothness regularization. SMURF specializes in estimating motion in consecutive frames, with checkpoints trained on KITTI, Sintel, and FlyingChairs.

**Doduo** is a method for finding dense correspondence across images trained without any human annotations on in-the-wild videos from Youtube-VOS (Xu et al., 2018). It uses photometric loss and a feature-metric variant using DINO (Caron et al., 2021) features. Doduo is not strictly unsupervised, as it uses off-the-shelf Mask2former segments (Cheng et al., 2022). For a fair comparison, we also report their numbers from an ablation training run without these masks.

**SEA-RAFT** is a supervised optical flow method that builds upon the original RAFT (Teed & Deng, 2020) by adding additional pretraining on TartanAir (Wang et al., 2020b), a novel mixture of Laplace loss and improve the initialization of the flow estimation.

### 4.2 BASELINES

We compare with the state-of-the-art supervised SEA-RAFT (Wang et al., 2024b) and unsupervised optical flow methods SMURF (Stone et al., 2021) and Doduo (Jiang et al., 2024).

### 4.3 COMPARISON TO SOTA METHODS

We compare with baselines on TAP-Vid DAVIS, using both VFG in Table 1 and CFG in Table 2. Our best performing models accept 512 resolution inputs and are evaluated with MM-40. On the VFG protocol, CWM with learned interventions significantly outperforms SMURF on all metrics. Further, our best performing model is able to outperform Doduo with supervised masks on all metrics and when the masks are removed, making Doduo fully unsupervised, the gap increases. On the CFG protocol, our best performing models are competitive with all baselines, regardless of supervision. Our model shows particularly strong performance on the AD metric, outperforming SMURF.

We show qualitative results in Figure 5. CWM is able to accurately track a point's movement through long frame gaps and complex dynamics. SMURF fails to accurately compute flow when there is large object or camera motion between frames. CWM qualitatively is more robust than Doduo in handling extreme cases, often accurately tracking points on the extremities of foreground objects or on rapidly shifting backgrounds.

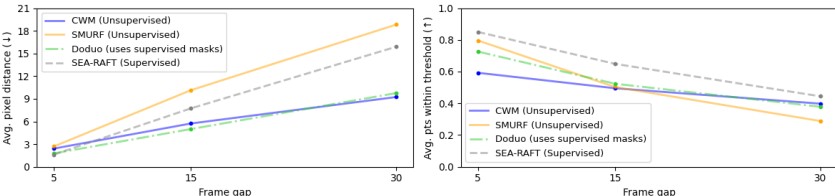

Figure 7: **Performance of optical flow methods as a function of frame gap.** We evaluate CWM, SMURF, Doduo, and SEA-RAFT on TAP-Vid DAVIS CFG with three choices of fixed frame gap (5, 15, and 30), comparing their performance as the amount of motion in each frame pair increases. Compared to SMURF and SEA-RAFT, CWM and Doduo are far more robust to larger frame gaps.

Table 3: **Analysis of CWM variations on TAP-Vid DAVIS VFG.** We compare our optimized counterfactual interventions with the fixed counterfactuals introduced in previous works (Bear et al., 2023). We demonstrate a clear improvement on all metrics, highlighting the need for bespoke, in-distribution counterfactual interventions.

| Resolution | Counterfactual | MM iters | AD↓ | MD↓ | $< \delta_{avg}^{x}$ ↑ |
|---|---|---|---|---|---|
| 256 | learned | 1 | 19.72 | 8.87 | 34.52 |
| 256 | learned | 40 | 14.63 | 5.84 | 42.62 |
| 512 | red square | 1 | 21.92 | 12.49 | 27.80 |
| 512 | green square | 1 | 17.66 | 8.22 | 34.90 |
| 512 | learned | 1 | 14.97 | 6.45 | 40.54 |
| 512 | learned | 40 | **12.45** | **4.62** | **47.50** |

## 4.4 ANALYSIS OF CWM DESIGN CHOICES

We present analysis across the input resolution of the RGB-conditioned predictor, $\Psi^{\text{RGB}}$, the number of masking iterations used for MM, and the form of the counterfactual intervention function, $\mathcal{C}$. The best-performing models use a 512 input resolution with MM-40.

By default, $\mathcal{C}$ allows for a variety of possible gaussian counterfactual interventions, with the Gaussian for each color channel optimized independently. We observe that with this structure and an initialization to random Gaussians, the three different-colored gaussians tend to converge to a similar shape and location (see Figure 6). We implement an initialization procedure so that $\mathcal{C}$ produces interventions with similar gaussians for each color channel. With overlapping gaussians of each color, these initial interventions look like "white bumps" (WBinit). Models trained with WBinit outperform all other (non-distilled) CWM-based models (see Tables 1, 2).

While our distilled model is relatively weak on the average distance (AD) metrics, especially in the high-motion VFG setting, on the $< \delta_{avg}^{x}$ metric (average points within a threshold) it is competitive with SMURF in both VFG and CFG, and outperforms all other CWM models in CFG. This demonstrates the effectivnes of our distillation procedure in the low frame gap setting.

We directly compared our optimized counterfactual interventions with the solid-color patches (Bear et al., 2023) and found that learned interventions perform better (see Table 3). This demonstrates not only that the CWM framework is highly effective at unsupervised motion estimation, but also that learning the counterfactual interventions is critical for good performance. We also show here (Table 3) that models with a larger input resolution outperform those with a smaller one, and that our multi-masking procedure significantly improves VFG metrics.

## 5 CONCLUSION

We demonstrate how to improve the performance generic CWM framework by learning to predict counterfactual interventions, and demonstrate the efficacy of this approach at estimating optical flow. Our approach takes an important first step towards optimizing counterfactual interventions for other visual structures like object segments and depth maps, while also improving upon state-of-the-art results for unsupervised optical flow estimation. Our findings indicate that CWM flow from learned counterfactual interventions is robust to various levels of object and camera motion compared to the existing SOTA baselines. In the future, we aim to extend our results to other downstream tasks. We plan to develop various differentiable counterfactual programs to extract higher-level visual structures such as segmentation, depth, keypoints, and dynamics, working our way toward a deeper learned understanding of the visual world.

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

## A APPENDIX

You may include other additional sections here.

