# OpenReview forum: "Learning Counterfactual Interventions for Self-Supervised Motion Estimation"
_ICLR.cc/2025/Conference — ICLR 2025 Conference Withdrawn Submission_

### Official Review · Reviewer_sthF · 2024-10-28

**Soundness:** 2
**Presentation:** 3
**Contribution:** 2
**Rating:** 3
**Confidence:** 2

**Summary:**

The paper explores a method to improve motion estimation in videos using a self-supervised approach. The authors address challenges in extracting usable information from visual representations without labeled data. They build on the Counterfactual World Modeling (CWM) paradigm, which uses a masked next-frame predictor to extract scene properties like optical flow.

**Strengths:**

1. The paper is well written and the approach is sound.
2. The method demonstrates performance improvements compared to the previous state-of-the-art.
3. The paper includes quantitative and qualitative comparisons, illustrating the strengths of the proposed method in various scenarios and frame gaps.

**Weaknesses:**

1. It seems to lack novelty, using a diffusion model to predict the next frame is too common. Firstly, the method proposed by the authors closely resembles the existing Counterfactual World Model. The counterfactual interventions proposed by the authors appear to utilize an alternative predictor to generate counterfactual predictions.
2. The overall amount of work appears too minimal.

**Questions:**

See weaknesses above. Authors can explain their core contributions to the task in the rebuttal. In determining the final score, I will take into consideration the opinions of the other reviewers.

---

### Official Review · Reviewer_YKwK · 2024-11-04

**Soundness:** 3
**Presentation:** 2
**Contribution:** 1
**Rating:** 5
**Confidence:** 3

**Summary:**

The authors propose to extend the counterfactual world models (CWM) by addressing the limitations in the hand-designed interventions. Specifically, hand-designed perturbations that are not found in the training distribution can cause spurious predictions and inconsistent motion estimates. To resolve this, the authors propose a paradigm to learn the interventions without using any additional supervisions. Quantitative results demonstrate superior performance against baselines.

**Strengths:**

[+] The work is well motivated by the limitation in hand-designed perturbation, since the model would intuitively struggle to predict something that it has never seen before.

[+] Figure 1 is clear and helps in illustrating what is CWM and how hand-designed perturbation may be limited.

[+] The proposed multi-mask inference approach is intuitive and effective according to the ablation study.

[+] Empirical evaluation shows superior performance on tracking across large temporal gaps.

**Weaknesses:**

[-] The contribution appears to be rather incremental. The counterfactual paradigm was established in previous works and remained mostly the same. The proposed architecture mainly served for learning a better "learned" intervention for probing motion information (tracking).

[-] Although the proposed auxiliary task for learning diffFLOW without supervision is interesting, I am not certain it is necessary and appears overly complicated. Would it be possible to use an off-the-shelf self-supervised model to produce correspondences for learning perturbations? This should improve training efficiency, reduce memory constraints, and provide a more explicit form of supervision.

[-] Figure 3 overlaps extensively with Figure 1 and Figure 2. Some separation / focus on particular parts would be more helpful.

[-] SMURF is an optical flow method and the comparison in Table 1 may not be entirely fair. As the authors acknowledged as well (L420), "this is more challenging than optical flow estimation.." Some discussion / additional baselines would be helpful.

[-] Minor errors - (L97): "showed that initially promising with this approach"

**Questions:**

Beyond the question that I raised in the weakness section, I am curious how well hand-crafted perturbations could work if these are not bright color patches that are obviously OOD, but something more in-distribution to the training data (e.g. randomly sampling spherical gaussians of jittered colors). High-level discussions (without experiments) would be sufficient here.

---

### Official Review · Reviewer_HK1k · 2024-11-05

**Soundness:** 3
**Presentation:** 3
**Contribution:** 3
**Rating:** 6
**Confidence:** 3

**Summary:**

This paper introduces a Counterfactual World Modeling framework that employs masked next-frame prediction to extract dynamic visual information in a counterfactual manner. A novel approach is proposed, where instead of relying on traditional fixed counterfactuals, the model learns to predict counterfactual interventions. To achieve this, the authors extend the conventional framework by parameterizing and optimizing optical flow extraction using a differentiable program called diffFLOW. By linking the outputs of diffFLOW to a flow-conditioned next-frame predictor and optimizing both jointly, the method ensures that the learned parameters of diffFLOW generate meaningful gradients for training counterfactual interventions. Extensive experiments demonstrate the effectiveness of this method in estimating optical flow.

**Strengths:**

1. The proposed framework enhances the generic CWM model by learning to predict counterfactual interventions, marking an advancement in the field.
2. The proposed method achieves state-of-the-art performance in motion estimation on real-world videos, demonstrating the approach's effectiveness even when applied to datasets without annotations.
3. The paper is well written, and it is easy to follow its main idea.

**Weaknesses:**

1. Some technique details and the motivation for the specific design are still unclear. According to Figure 3, the model requires an initial counterfactual p, which is then used to further learn the counterfactual intervention through the optimized MLP. If I understand correctly, the training process requires an initial $p$. How does the initial position of p affect the final motion estimation results? How should $p$ be chosen for different datasets? Does the initial color of p have any impact on the optimization results? These kinds of questions introduce significant uncertainty in how to obtain good results using the existing framework. It would be beneficial to include some ablation studies on this aspect.

2. In the table1 and table2, different models are optimized using different datasets. Why not standardize the datasets used across the different methods? This seems to be a more reasonable and fair setup for comparing the performance of different methods.

**Questions:**

1. It would be beneficial to include some ablation studies about the choice of the initial $p$.
2. It would be beneficial to provide results for different methods trained on the same dataset, whether in a supervised or unsupervised manner. Alternatively, at the very least, a reasonable justification for the current experimental setup should be given.

**Details Of Ethics Concerns:**

This paper introduces a novel counterfactual world modeling paradigm for self-supervised motion estimation. All of the experiments focus on computer vision tasks; therefore, I believe no ethical review is required.

---

### Official Review · Reviewer_XJqk · 2024-11-06

**Soundness:** 3
**Presentation:** 3
**Contribution:** 3
**Rating:** 6
**Confidence:** 4

**Summary:**

This paper tackles a key issue in self-supervised learning from visual data: effectively extracting useful information from the representations learned during pretraining, such as motion estimation. The authors build on the counterfactual world modelling paradigm, which uses a masked next-frame predictor to facilitate the extraction of key visual features, such as optical flow, segments, and depth. They specifically target motion estimation for real-world videos, providing a promising step forward in unsupervised visual feature extraction. The key novelty of this paper lies in enhancing self-supervised motion estimation through optimized counterfactual interventions within the Counterfactual World Modeling (CWM) paradigm. Traditional CWM methods use fixed interventions, such as colored patches, which can be inconsistent and noisy in their predictions. This paper introduces a differentiable, learned counterfactual function, termed "diffFLOW," that leverages Gaussian interventions for improved optical flow extraction.

**Strengths:**

- The paper is easy to follow and well-presented
- The use of a learned counterfactual intervention function is unique and provides a tighter, more effective alignment between pretext tasks and motion estimation.
- The method outperforms current unsupervised motion estimation benchmarks, especially on datasets with challenging frame gaps and motion dynamics.
- By reducing reliance on labeled data, this approach has high potential for scalability across large video datasets.

**Weaknesses:**

- The multi-mask inference technique, while improving accuracy, increases inference time, potentially limiting real-time applications.
- The paper primarily focuses on optical flow and does not extensively demonstrate results for other visual properties like depth and segmentation, which are mentioned as future directions.


---

Typo
L083  from from

**Questions:**

- How does the model perform under different types of motion, such as fast-moving objects, complex rotations, or occlusions? Are there particular types of motion where diffFLOW struggles?
- How does this method compare with other state-of-the-art models, such as SMURF and SEA-RAFT, regarding accuracy and efficiency across varying frame gaps?
- L288 to L293 Have you done the ablation with solid-coloured squares?

---

### Official Review · Reviewer_fMDP · 2024-11-09

**Soundness:** 2
**Presentation:** 2
**Contribution:** 3
**Rating:** 5
**Confidence:** 3

**Summary:**

This paper proposes a self-supervised estimation of optical flow by learning counterfactual interventions instead of hand-designed counterfactual interventions. Specifically, It re-formulates the motion extraction procedure to make it a parameterized differentiable function and introduces the functional form of a sum of colored Gaussians as a natural intervention class. It claims that the proposed method achieve state-of-the-art performance on TAP-Vid DAVIS—VFG and TAP-Vid DAVIS—CFG. Generally, the idea makes sense to me.

**Strengths:**

1. The idea of learning counterfactual Interventions instead of hand-design counterfactual interventions is interesting and novel.
2. Jointly learning counterfactual Interventions and a counterfactual motion prediction to make the system end-to-end is a good design.

**Weaknesses:**

1. Related wok is quite rough and lack comparisons to the proposed method. Especially, the second paragraph of the related work is less details.
2. The propose method lack details and further explanation:
1) In L214, it is not clear why the predicted pixel location pˆ2 can be retrieved by finding the peak in the difference image?
2) In L239, "While sometimes effective, a bright colored patch is out of domain for the base predictor." Is there any reference or analysis to support this statement?
3) In Figure 3, I did not find p1 and Ψflow which makes it difficult for me to understand.
4) Does diffFLOW : (I1 , I2 , p1 ) generate flow vector for each pixel? It processed pixel by pixel for the image?
5) In L306, " the first frame RGB input I and predicts the next frame Iˆ , conditioned on the flow input Fˆ." Specifically, how the model utilizes the Flow F' as condition?
6) Why The final MM prediction is the peak in the average delta image? Any theory analysis?
7) Section 3.4 lack of details for how to distill it into an architecture purpose-built for optical flow estimation.

3. Experiment:
1) It claims that the performance is state-of-the-art on TAP-Vid DAVIS CFG in Table 2 but actually it does not beat SMURF for all metrics.
2) From Table 1 and Table 2, the distilled results are not good enough as MM.
3) As stated in the paper, MM-40 has good results but makes inference expensive. Any comparisons to other SOTA methods on the computation cost?

**Questions:**

See the weakness.

---

### Author Response · Authors · 2024-12-02
**General Response**

We thank the reviewers for their feedback, which we will use to improve the paper and resubmit to a future venue. We offer some brief clarifications to re-emphasize the main points of our work.

Our main goal is to investigate how far we can push the performance of the counterfactual world modeling (CWM) paradigm by leveraging its core design principle: the tight coupling between the pretext task (sparse-RGB conditioned next frame prediction) and the counterfactual structure extraction procedures that allow for zero-shot extraction of optical flow or object segments. The key idea is to take advantage of this tight coupling and improve the quality of extractions (in our case optical flow) by parameterizing and optimizing the counterfactual procedures without any supervision from human labels or heuristics like off-the-shelf model pseudo-labels that are inherently limited. This is why we use another next frame prediction task, but conditioned on sparse flow, to obtain gradients for optimizing the counterfactual procedure for extracting flow.

To properly test the potential of this paradigm, we train on large-scale diverse videos from Kinetics and use generic ViT-based architectures, rather than synthetic datasets or small-scale real data and highly specialized architectures with complex loss formulations. The point of our baseline comparisons is to show that it is possible for a generic approach like CWM, from which multiple visual structures can be extracted, to outperform the specialized self-supervised baselines, which has not been demonstrated before. Regarding inference speed and real-time applications, we have already shown the potential for distilling the representation from CWM into a smaller and faster model, and further engineering efforts (quantity and density of labels used for distillation) will yield improved distilled models.

---

### Note · Authors · 2024-11-14

I have read and agree with the venue's withdrawal policy on behalf of myself and my co-authors.

---

> ### Note · Program_Chairs · 2024-12-02
>
> **Comment:**
>
> Withdrawal reversed temporarily.
>
> **Revert Withdrawal Confirmation:**
>
> We approve the reversion of withdrawn submission.

---

### Note · Authors · 2024-12-02

**Comment:**

Requested by authors.

**Withdrawal Confirmation:**

I have read and agree with the venue's withdrawal policy on behalf of myself and my co-authors.